# A new surgical tape with a mesh designed to prevent skin tears and reduce pain during tape removal

**Naoaki Rikihisa**[1,2]*, **Nobuyuki Mitsukawa**[3], **Hiroaki Rikhisa**[4], **Masayuki Nakao**[5]

**1** Oyumio Central Hospital, Chiba, Japan, **2** Tokyo Medical Tape, Yokohama, Japan, **3** Department of Plastic, Reconstructive, and Aesthetic Surgery, Chiba University Graduate School of Medicine, Chiba, Japan, **4** Research and Development Division, Furukawa Electric, Yokohama, Japan, **5** School of Engineering, The University of Tokyo, Tokyo, Japan

* rikihisa@faculty.chiba-u.jp

**Data Availability Statement:** All relevant data are within the manuscript.

**Funding:** This study was supported by Entrepreneurs Program of New Energy and Industrial Technology Development Organization

## Abstract

We devised a surgical tape that prevents skin tears while maintaining adhesive strength. Under the assumption that microscopic damage to the skin is reflected in pain felt on the skin, we statistically analyzed skin pain when the tape was peeled off to show the skin protection effect of the mesh on the new tape. This tape has a three-layer structure consisting of a tape substrate, adhesive, and mesh. When the tape is applied to the skin, a mesh is located between the adhesive and the skin. The adhesive contacts the skin through the mesh holes and fixes the substrate to the skin; it does not come into contact with the skin at the mesh body; therefore, the adhesive-skin contact area is reduced. In this experiment, we used surgical tape with and without mesh. At 8 hours after the application of each tape to the forearm of five adult males, it was removed. All tapes were peeled off while maintaining an angle of approximately 120° between the skin and tape substrate. For the tape with mesh, the tape substrate was peeled off in two ways: peeling off the substrate together with the mesh and peeling off the substrate, leaving the mesh on the skin. A perception and pain quantification analyzer (Pain Vision™) was used to quantify pain. The data were compared and examined statistically (Friedman's test and Wilcoxon's coded rank test). The least pain was experienced while peeling off the tape substrate, leaving the mesh on the skin. There was a significant difference in pain levels among the three tape removal methods. There was also a significant difference between the two peeling methods in the experimental group. The skin protection effect of the mesh reduced pain when the surgical tape was removed.

## Introduction

The skin of older adults suffers from collagen degradation, loss, and dryness. When the surgical tape is removed, fragile skin bends considerably, and damage can be seen in the form of skin tears, where the skin is peeled off between the dermis and subcutaneous tissue, which is likely to occur [1–4]. In nursing education, efforts to avoid skin tears have mainly been made,

(22100741-0) in Japan. The funders had no role in the study design, data collection and analysis, decision to publish, or preparation of the manuscript. URL of the funder website is https://www.nedo.go.jp/.

**Competing interests:** We have read the journal's policy and the authors of this manuscript have the following competing interests: Hiroaki Rikhisa is a holder of the patents related to the surgical tape with the mesh. This study supported by Entrepreneurs Program of New Energy and Industrial Technology Development Organization (22100741-0). This does not alter our adherence to PLOS ONE policies on sharing data and materials.

and silicone-based adhesive tapes have been commercialized to meet this need. However, skin tears cannot be prevented entirely [3, 4].

Therefore, we tried creating a surgical tape that has strong adhesiveness but does not damage the skin when removed. In a previous experiment, we demonstrated the skin protective effect of our new tape using the mechanical calculation formula of the beam structure and the tape application and peeling test (*in vitro*) on the eggshell membrane of a chicken egg (the thin membrane of boiled eggs) were regarded as the atrophied skin of older adults. The skin protection effect of mesh was demonstrated with a thin protein film, and the mechanism of mesh action was discussed using architectural engineering techniques [5]. In this study, we applied a new tape to the skin of five healthy male adults; pain level was measured and analyzed when the tape was removed (in vivo). The skin-protective effect of the new tape was indirectly verified. A tape manufacturer combined mesh with the medical tape used to secure dialysis routes in hospitals to create the experimental tape. We attempted to clarify the advantages and disadvantages of mesh by adding no new components of medical tape other than mesh in this experiment.

## Materials and methods

### Structure and function of the new tape

**Structure of the new tape.** The new tape had a three-layer structure consisting of a tape substrate, adhesive, and the mesh. When this tape was applied to the skin, a mesh was present between the adhesive and skin. The adhesive contacted the skin through the mesh holes, fixing the tape substrate to the skin. However, the adhesive did not come into contact with the skin at the mesh body; hence, the adhesive-skin contact surface was divided by the mesh (Fig 1).

The mesh was made of olefin using the printing method. The openings of the mesh were pentagonal, with a line thickness of 250 μm, width of 750 μm, and open area ratio of 76% (Fig 2). Kino white (Nitoms, Inc., Tokyo, Japan) was used as the tape base material and the adhesive of the experimental tape. Kino white is a highly adhesive medical tape composed of non-woven paper and acrylic adhesive, and is used in dialysis hospitals to fix tubes to the patient's skin. The experimental tape was made by a tape manufacturer (Cosmotec Co., Ltd.) by combining Kino white and olefin mesh.

**How to use the mesh when peeling off the tape.** The mesh should be held (A side) with one finger (such as pressing the mesh toward the patient's skin), and the tape substrate should be pinched with the other finger and slowly peeled off. Because the B side of the mesh was fixed to the skin using tape, a downward force on the skin was applied to the mesh from A and B (Fig 3). Subsequently, this downward force was transmitted to the skin through the mesh, preventing it from sagging because the tape pulled it up (Fig 3).

### The tape-peeling test

After shaving both forearms of the 5 adult males with clippers, control surgical tape without mesh and the new surgical tape with mesh were applied. We used Kino white (width, 2.5 cm; length, 5 cm; Nitoms, Inc., Tokyo, Japan) as the control tape. The experimental tape used a 2.5 × 5-cm Kino white and a 2.5 × 6-cm mesh. The mean adhesive strength of the control tape and experimental tape was 4.43 N/25 mm and 1.42 N/25 mm, respectively. Adhesion of the tapes was measured according to testing methods of pressure-sensitive adhesive tapes and sheets in Japanese Industrial Standards. Two pieces of control tape and two pieces of experimental tape were applied to the right forearm. In addition, four pieces of experimental tape were attached to the left forearm. At 8 hours after the application, a tape-peeling test was performed.

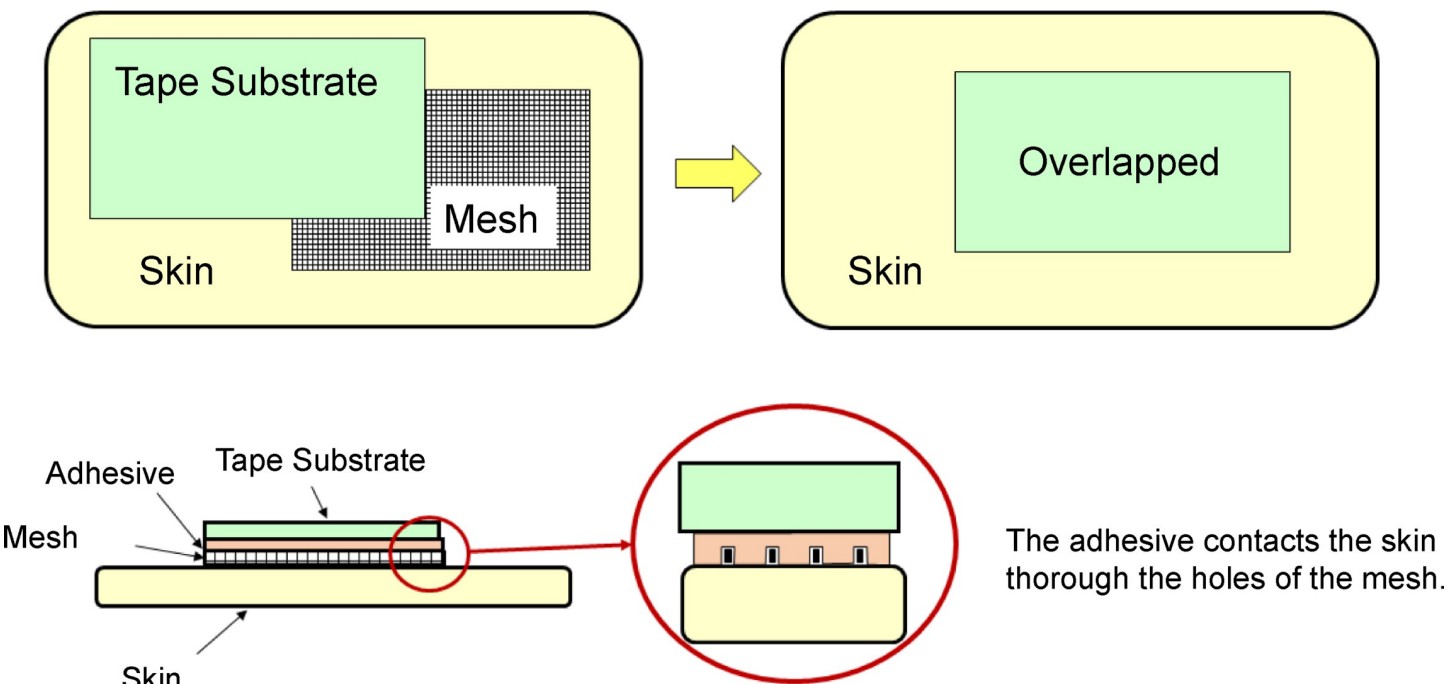

**Fig 1. Structure of the new surgical tape with mesh.** A mesh was placed under the tape substrate. As the adhesive contacted the skin through the mesh holes, the entire surgical tape could be fixed to the skin. The adhesive surface was divided by the mesh because the adhesive did not come into contact with the skin around the area where the mesh was present.

Pain Vision™ (NIPRO Co., Ltd., Osaka, Japan) (Fig 4) was used to quantify and record the tape-peeling stimulus [6]. Pain Vision™ is an analyzer that calculates and displays the "degree of pain" from the current perception threshold (CPT), which is the minimum amount of electrical stimulation felt by a study participant, and the pain-compatible current (PCC), which corresponds to the perceived pain. This device gradually increased the stimulation current from 0 μA and measured the CPT when the patient felt some stimulation. As the current was increased, this device measured the PCC when "stimulation of the electrode site became equivalent to the patient's perceived pain." Subsequently, the degree of pain was calculated using the following arithmetic expression [2]:

Degree of pain = $100 \times (PCC - CPT)/CPT$, where PCC is the pain-compatible current and CPT is the current perception threshold.

The electrodes of the measuring device were attached to the participants' left forearm, and the participants pressed a button with their right hand to record the stimulus. The measurement items were as follows: (i) Skin irritation was quantified when removing the tape without mesh (control group); the number of measurements was two. (ii) Skin irritation was quantified when the tape with mesh (experimental group 1) was peeled off while leaving the mesh on the skin; three measurements were performed. (iii) Skin irritation was quantified when the tape with mesh (experimental group 2) was peeled off together with the mesh; three measurements were performed.

The examiner removed the tape while the participants were instructed to close their eyes. At this time, the participants were unaware of the tape being removed. Eight tapes were removed in random order by the same examiner. When the tape was peeled off, the tape substrate was made to form an angle of approximately 120° with the skin, and consideration was given to ensure that the tape was peeled off at the same speed. The participants were allowed to

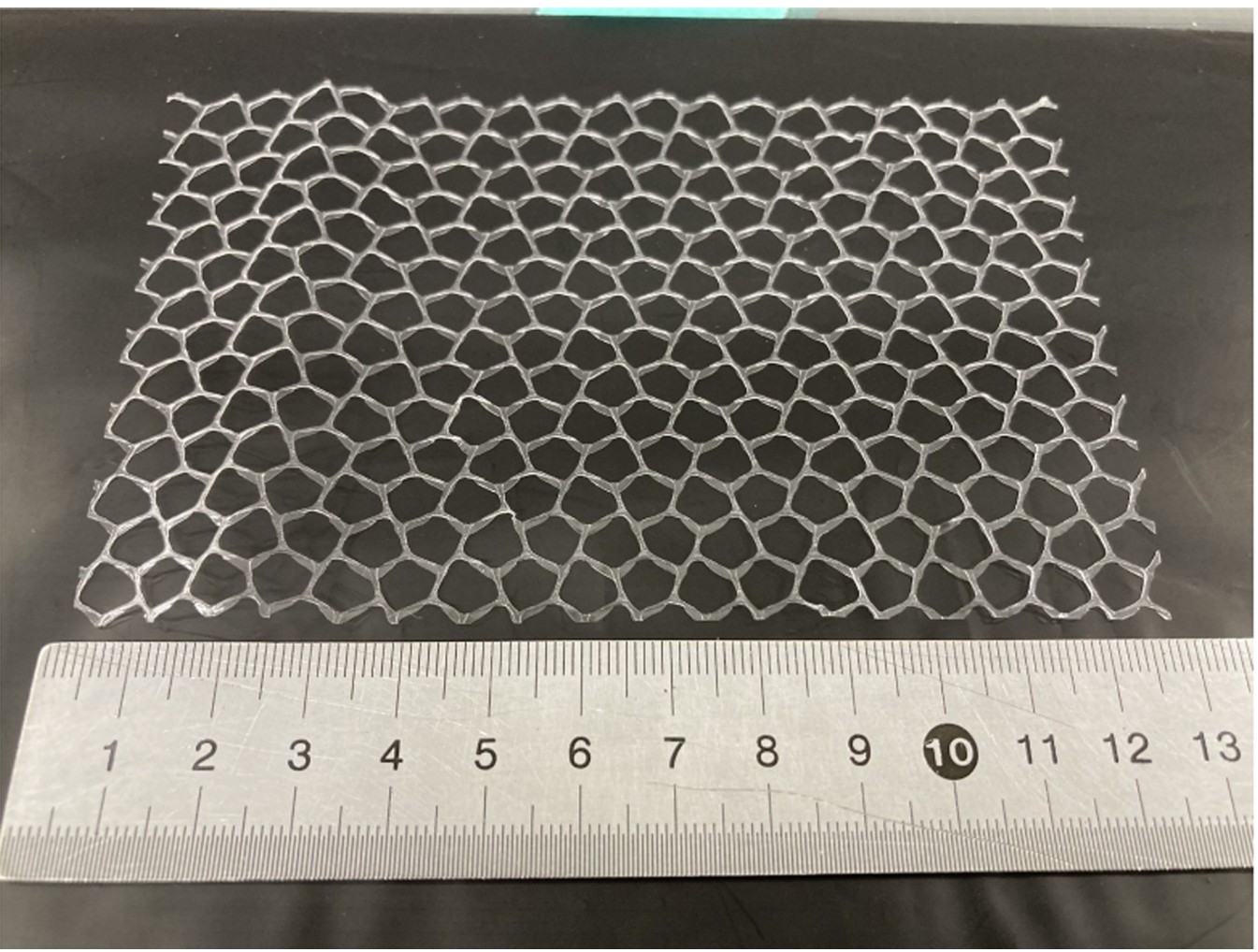

**Fig 2. The appearance of the mesh.** The mesh was made of olefin using the printing method. The openings of the mesh were pentagonal, with a line thickness of 250 μm, width of 750 μm, and open area ratio of 76%.

live their daily lives while the tape was applied. The participants' forearms were photographed and documented after tape application and removal. A mock experiment was conducted prior to the study to familiarize the participants with the pain measurement procedure using the device (Pain Vision™).

### Recording the state of the skin when the tape with mesh was peeled off

The skin was filmed to visually confirm the function of the mesh attached to the surgical tape, and the new tape was peeled off using the following two methods: first, the tape with mesh was peeled off while leaving the mesh on the skin; second, the tape with mesh was peeled off along with the mesh. In both methods, the tape was removed while maintaining an angle of approximately 120° between the skin and tape.

### Statistical analyses

Statistical analyses were performed using HAD (https://osf.io/32cyp/#), which is a graphical user interface-based free software program with various functions for basic statistical and

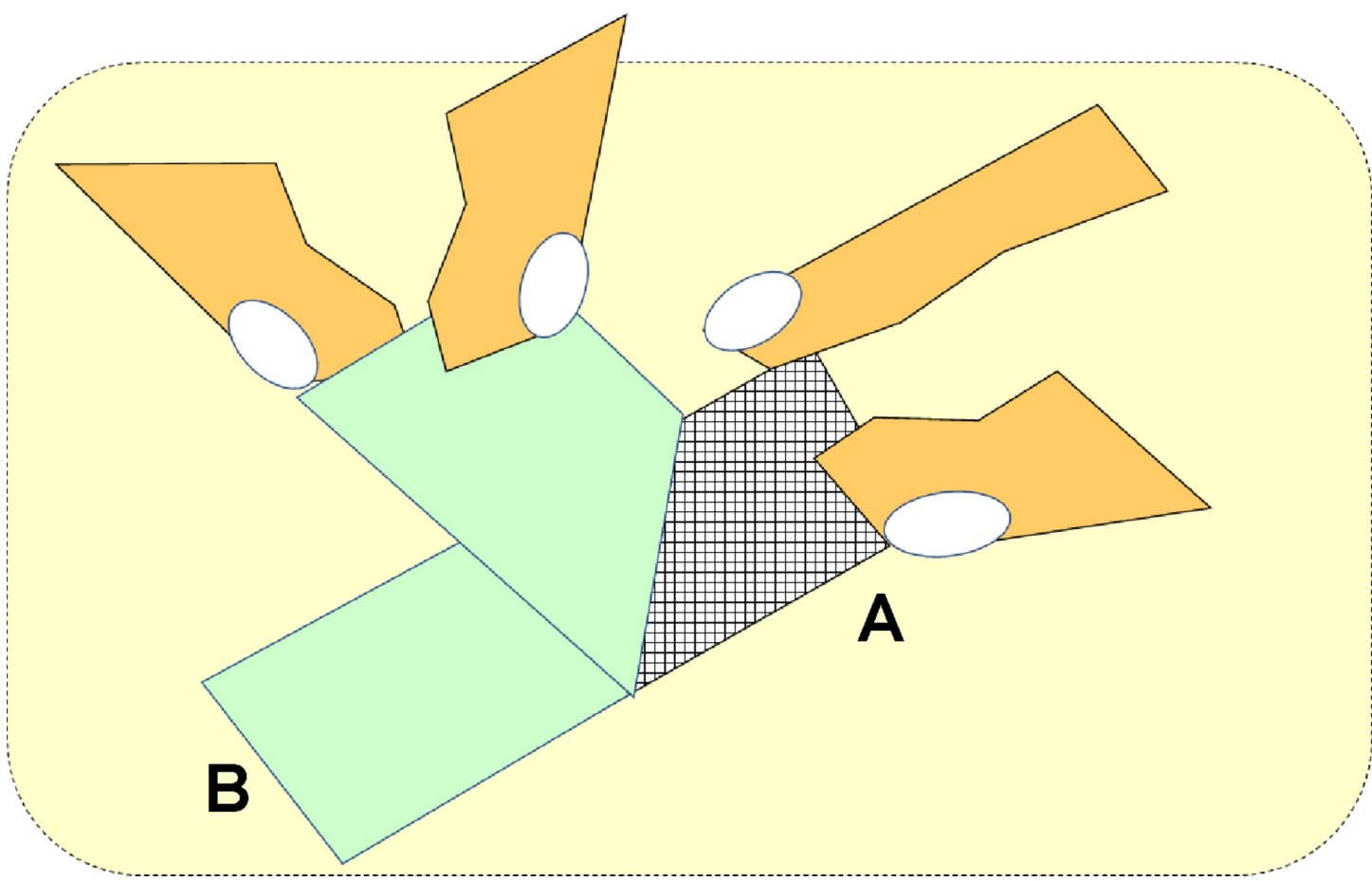

**Fig 3. How to use the mesh when removing the tape.** While pressing the mesh (A side) with one finger, pinch the tape base with the other finger and slowly peel it off. The downward force applied from A and B prevents skin deformation.

multivariate analyses [7]. Data distributions and box plots were created for each group. Non-parametric tests were performed because the population did not follow a normal distribution. The following test methods were adopted to determine whether there was a difference in the distribution of two or three populations for the corresponding samples. Friedman's test was performed for the three corresponding groups (control and experimental groups), and Wilcoxon's coded rank test was performed for the two corresponding groups (experimental groups). The significance level was set at $p < 0.05$, and two-sided 95% confidence intervals were set for each test.

## Ethical considerations

The ethics committee of Oyumino Central Hospital examined and approved the outline of the study, methods, purpose of tape application and peeling, possible disadvantages, and privacy protection. Prior to the study, the participants were informed verbally and in writing about the purpose, content, disadvantages, and privacy policy of this study; participation was voluntary and could be withdrawn during the study. Written consent was obtained from the study participants. Patients with skin diseases, allergies, and weak skin were excluded.

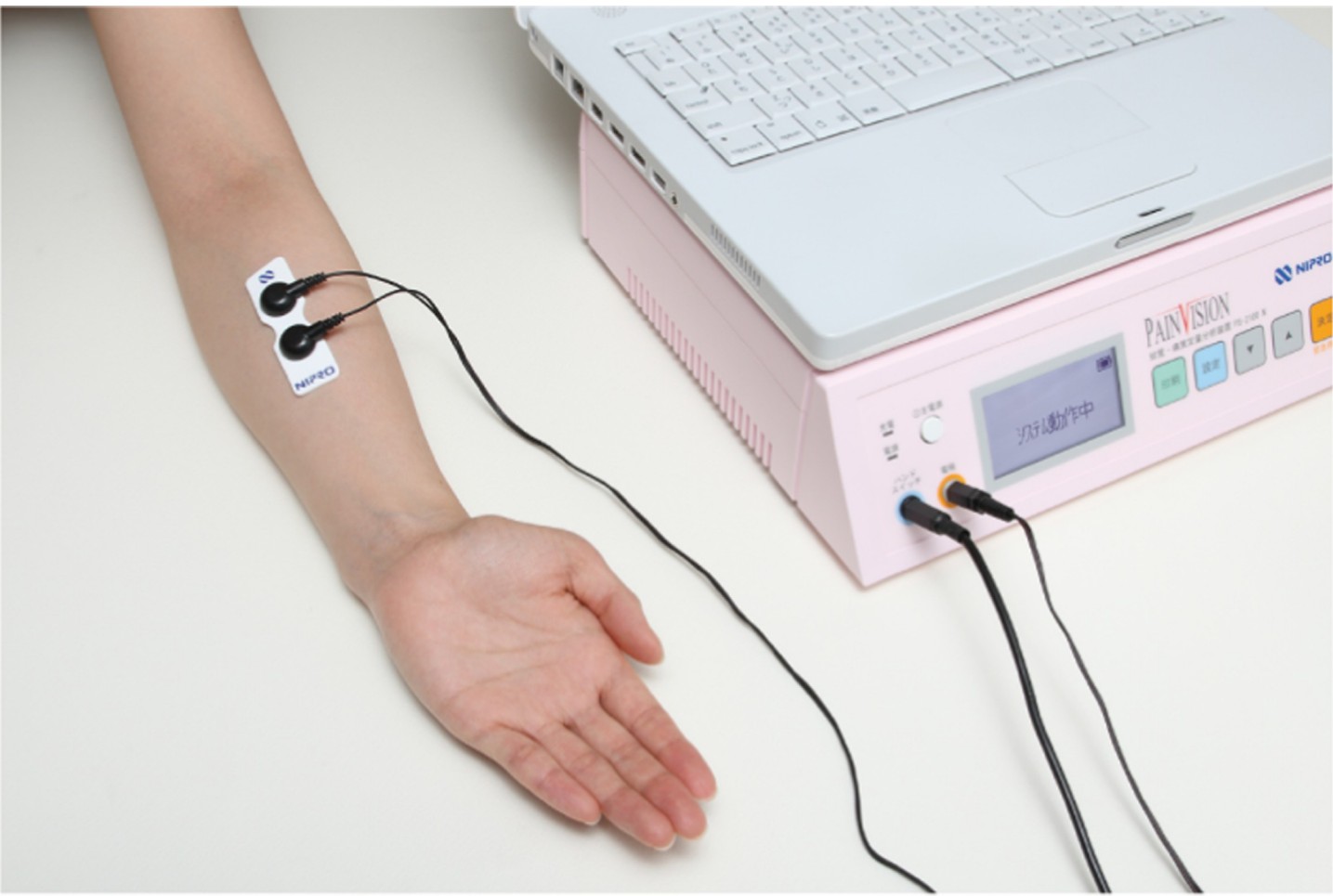

**Fig 4. The appearance of pain Vision™.** Pain Vision™ is a somatosensory-evoked nerve electrical stimulator instrument that shows the degree of pain. This device uses a unique waveform and current frequency to determine the degree of pain. There are no complicated operations, and one measurement could be performed in approximately 3 min (https://med.nipro.co.jp/med_eq_category_detail?id=a1U1000000b536NEAQ; converted from this website).

## Results

Fig 5 and S1 and S2 Videos show the state of the skin when the tape with mesh was peeled off. The deflection of the skin when the mesh was suppressed and only the substrate was peeled off (Fig 5, right, S2 Video) was smaller than its deflection when the mesh and substrate were peeled off together (Fig 5, left, S1 Video). S1 and S2 Videos are played at 0.25× speed. S1 Video shows the skin being pulled up gradually by the tape; after that, it was peeled off from the tape and returned to its original position. As shown in S2 Video, the amount of skin up-deformation was small without fluctuation because the mesh prevented the skin from being pulled up by the tape.

Data on the degree of pain were not normally distributed for each of the three types of tape peeling. Fig 6 shows box plots for the degree of pain among the three types of tape peeling. Fig 7 presents the results of Friedman's test for the control and experimental groups; a significant difference was observed ($p < 0.001$, $\eta^2 = 0.578$). Fig 8 presents the results of Wilcoxon's coded rank test between the experimental groups. Again, there was a significant difference between the two groups ($p = 0.03$, $r = 0.571$). No adverse events, such as dermatitis, were observed during the study.

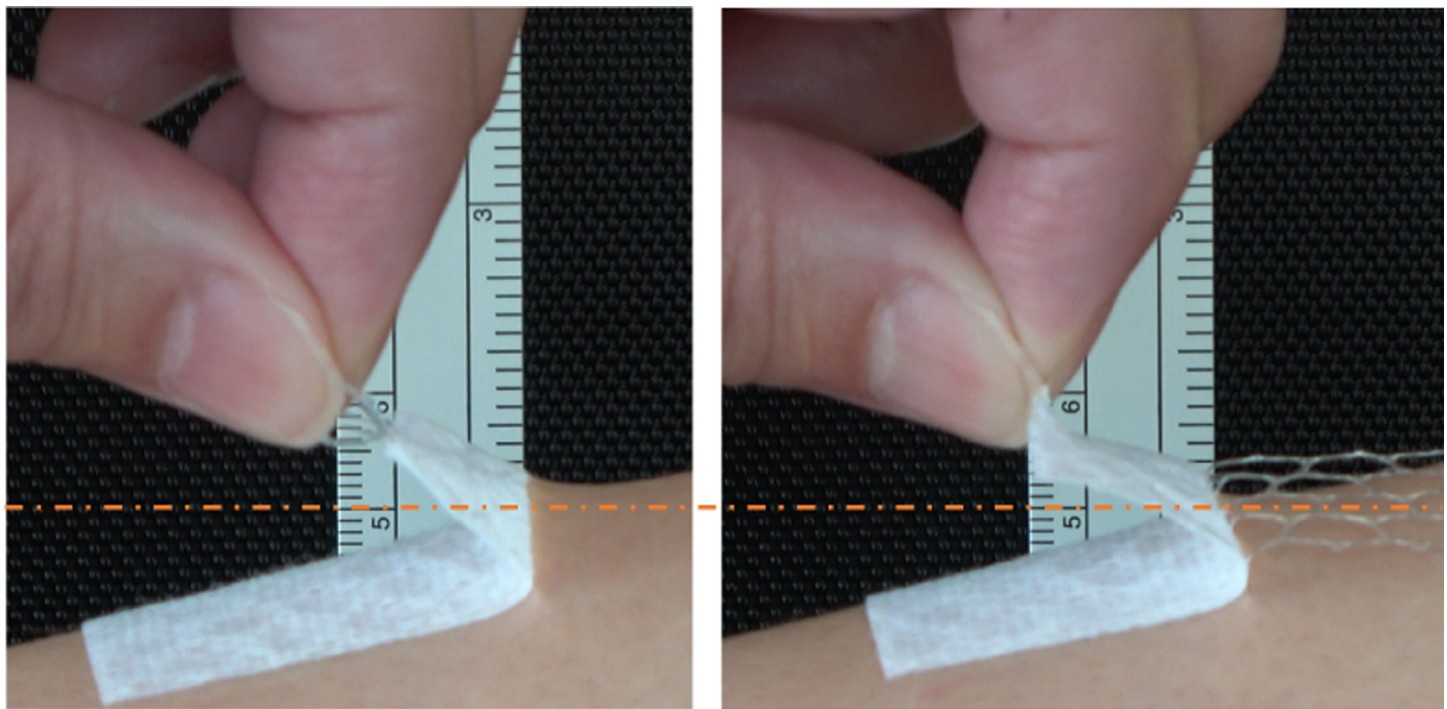

**Fig 5. The state of the skin when the tape with mesh is peeled off.** The deflection of the skin when the mesh was suppressed and only the substrate was peeled off (right) was smaller than its deflection when the mesh and substrate were peeled off together (left). In both methods, the tape was removed while maintaining an angle of approximately 120° between the skin and tape. An orange dashed line indicating the amount of skin deformation is added to the same standard photograph.

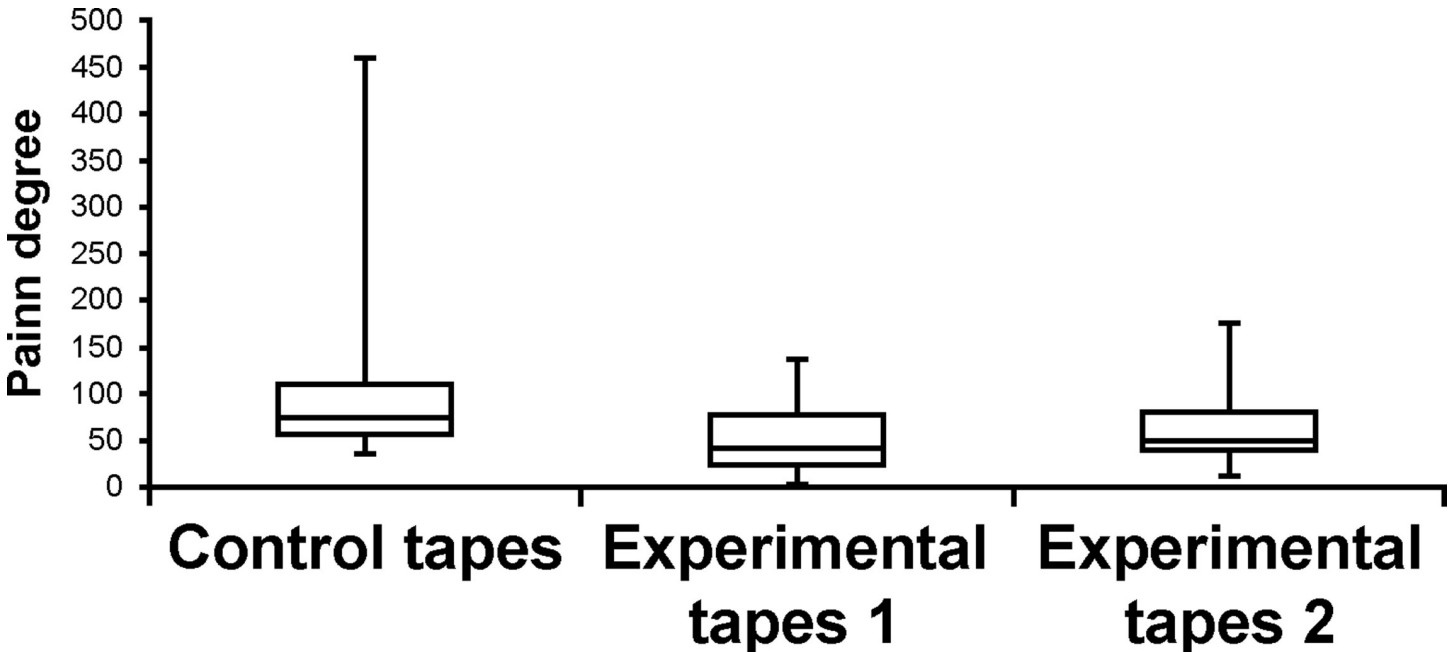

**Fig 6. Box plots for the degree of pain when the tape was peeled off among three types of peeling.** Control tapes: surgical tapes without the mesh. Experimental tapes: surgical tapes with the mesh. Experimental tape 1: the tape was peeled off while leaving the mesh on the skin. Experimental tape 2: the tape was peeled off together with the mesh from the skin.

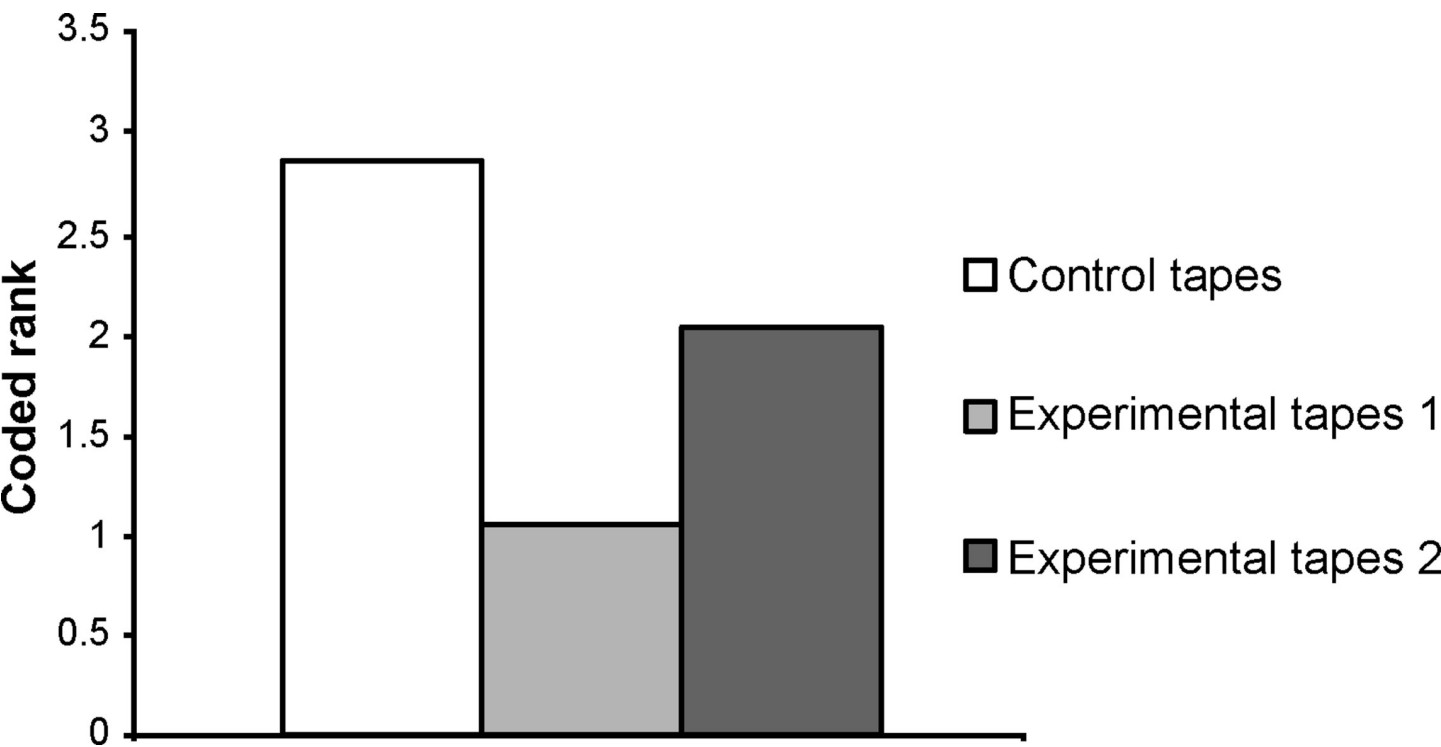

**Fig 7. The result of Friedman's test for the degree of pain when peeling off the tape in the control and experimental groups.** A significant difference was found (p <0.001). Control tapes: surgical tapes without the mesh. Experimental tapes: surgical tapes with the mesh. Experimental tape 1: the tape was peeled off while leaving the mesh on the skin. Experimental tape 2: the tape was peeled off together with the mesh from the skin.

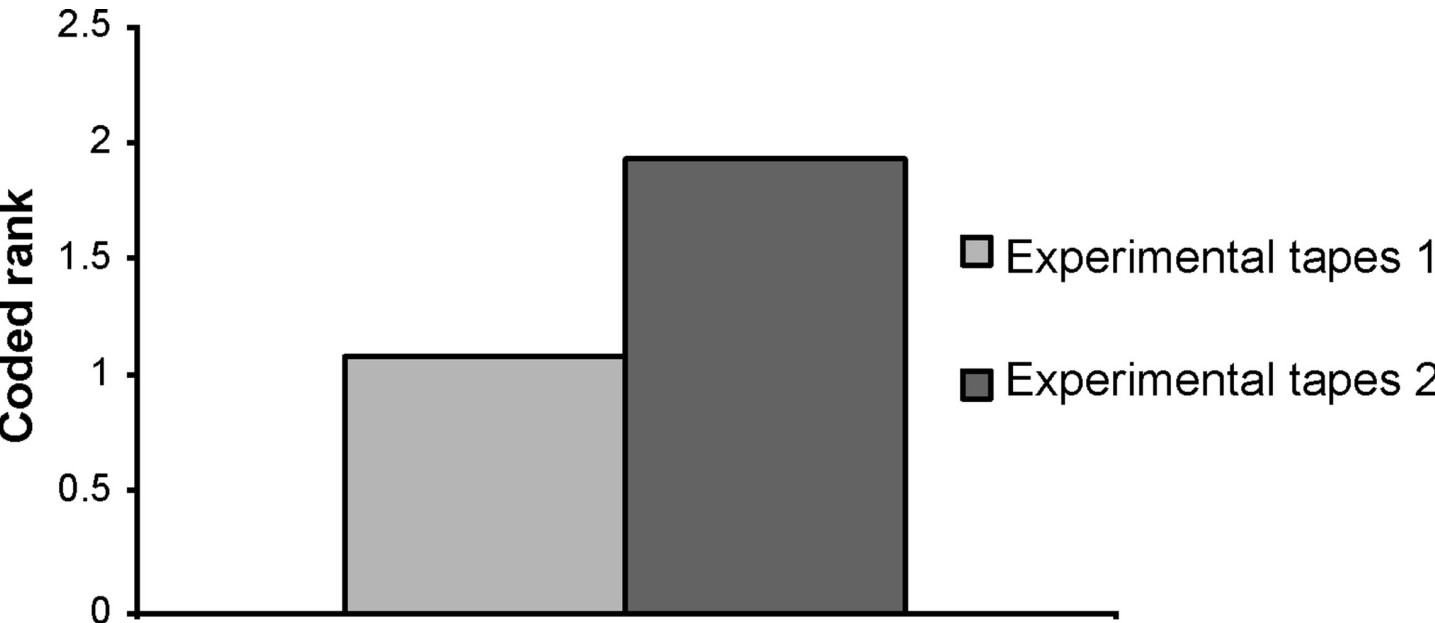

**Fig 8. The result of Wilcoxon's coded rank test for the degree of pain when peeling off the tape in the experimental groups.** A significant difference was found (p = 0.03). Control tapes: surgical tapes without the mesh. Experimental tapes: surgical tapes with the mesh. Experimental tape 1: the tape was peeled off while leaving the mesh on the skin. Experimental tape 2: the tape was peeled off together with the mesh from the skin.

## Discussion

When the tape was peeled off while leaving the mesh on the skin (experimental tape 1), the pain felt on the skin was significantly minimized (Figs 7 and 8). Pain Vision™ (NIPRO Co., Ltd., Osaka, Japan) (Fig 4) was used to quantify and record the tape-peeling stimulus [6].

The tape with mesh in the experimental groups had a lower contact area with the adhesive than the control tape with no mesh. The average adhesive strength of the control tape and experimental tape was 4.43 N/25 mm and 1.42 N/25 mm, respectively. Therefore, the pain felt while peeling off the tapes was reduced in the experimental group, as compared with that in the control group. A comparison between the experimental groups showed that the degree of pain differed, depending on how the tape was removed, even if the tape had the same mesh and adhesive area as the tape substrate.

Although, cutaneous pain is a very familiar phenomenon, the exact mechanisms by which pain arises are still under debate, physiologically and anatomically. Nevertheless, pain is caused by injury or stress to the body. Therefore, it is reasonable to assume that the pain sensation was reduced by reducing the stress on the skin when the tape was removed.

In a previous study, the eggshell membrane of boiled eggs did not peel off when the tape was peeled off, leaving the mesh on the egg surface by holding it with a finger; this indicates that the load on the eggshell membrane (atrophied skin model) during tape peeling was reduced. In other words, it was directly shown that the new tape with mesh could be peeled off from the skin while avoiding skin damage. Otherwise, the eggshell membrane broke when the tape was removed without holding down the mesh; it was observed that it adhered to the egg-shell membrane via the mesh holes [5].

The reduction effect of the mesh on the sensation of skin pain during tape peeling shown in this experiment complements the results of the previous experiment. The mechanism through which the mesh reduces skin damage and pain during tape stripping is thought to be mainly due to the following two functions; when the tape was peeled off, the skin directly under the mesh body could be pressed downward by the mesh (Fig 5), and by dividing the adhesive surface with a mesh, a continuous upward force was not applied when the tape was peeled off. Furthermore, when the substrate was peeled off while suppressing the mesh, the upward deflection of the skin was reduced. S1 and S2 Videos also show that the upward deflection of the skin was suppressed by leaving the mesh on the skin.

The second function of the mesh was discussed in a previous report using a mechanical calculation formula for the beam structure [8]. Assuming that Young's modulus is E, the geometrical moment of inertia is I, and the bending moment is M, the differential equation shown in Numerical Formula 1 holds for the deflection of beam V, which is the displacement in the vertical direction from the central axis $Y_0$ of the beam (S1 Fig).

$$\text{numerical formula 1;} \quad \frac{d^2v}{dx^2} = -\frac{M}{EI}$$

The method for calculating the deflection of a beam is the basis of mechanical design, and its concept and method have been established. In this discussion, the force applied to the beam corresponds to the force that pulls the skin upward, and the length of the beam corresponds to the length of the tape that adheres to the skin. The deflection of the beam corresponded to the deflection of the skin. In general, when applying a load to a beam or planar object, it is better to apply upward and downward forces alternately than to continuously apply forces in the same direction (e.g., gravity) to reduce the deflection of the beam. In the case of the new surgical tape, it is possible to apply force to the skin through the mesh from the fingertips that hold the mesh down so that the skin deflection can be reduced [8].

The suspension bridge is considered the bridge type that enables the longest span length among the existing bridge types, and the Golden Gate Bridge, 2737 m long, in San Francisco is a good example [8]. One of the characteristics of the suspension bridge structure is that multiple "suspension materials" lift the bridge at multiple points. Upward forces are applied to the bridge from multiple points, and these multiple upward forces resist gravity applied continuously downward. Our mesh corresponds to the suspension bridge's multiple "suspension materials," and the gravity on the bridge corresponds to the upward force of tape on the skin. Each works to avoid deformation of the patient's skin or the main bridge girders by shortening the moment that causes deformation [8].

A beneficial aspect of Pain Vision™ is that the pain sensation can be expressed in terms of current values. Therefore, the degree of pain was correlated with the commonly used visual analog pain scale. Although, defining the degree of pain as a completely objective evaluation is difficult, it is more accurate than a visual analog scale. In addition, it can sharply indicate the effect of pain treatment [8].

A clinical problem regarding the degree of pain is that pain is impossible to compare among multiple participants (such as Mr. A, Mr. B, and Mr. C) using this numerical value. Indeed, there was also a large variation in the degree of pain with the control tape-peeling stimulus (minimum, 35; maximum, 460; median, 75.5; average, 143.8), as shown in Fig 6. However, the degree of pain is beneficial for comparing the pain felt by the same participant, such as the pain felt on Mr. A's right arm and the pain felt on his left arm. Furthermore, electrical stimulation of Pain Vision™ is a sensation that is very similar to pain in the skin when the tape is removed. In addition, instead of judging the pain in the right and left arms by comparing them, the participants directly described the current pain by comparing it with the adjusted electrical stimulation. Consequently, these methods that quantify pain are advantageous for the statistical analyses of pain relief, such as those conducted in this study.

The skin of older adults degrades as a result of decreased collagen and dry skin; hence, when the surgical tape is removed, the skin considerably bends and is peeled off between the dermis and subcutaneous tissue, often leading to skin tears [1–4]. In addition, even if the adhesive strength of the surgical tape is weak immediately after it is applied to the skin, the adhesive strength increases as the contact area increases and the adhesive spreads over the uneven surface of the skin [9]. Therefore, nurses and caregivers must select surgical tapes according to the patient's underlying disease and skin condition and apply and remove them appropriately [10].

Surgical tapes are used to fix sanitary materials and equipment necessary for maintaining the health and life of a patient's skin. The fixation should not come off carelessly; hence, the surgical tape must have high adhesive strength. The function of not causing skin tears when peeled off while having strong adhesion is a universal demand. Moreover, the cheaper the surgical tape, the greater its merit. However, it is difficult to meet this universal function and the required cost because adhesives such as silicon adhesives bear two trade-off functions [4]. We attempted to solve this problem by dividing the conflicting functions into two applications. The function of each application can be enhanced by ensuring strong adhesion and using a mesh to protect the skin. It is a simple and common way to achieve two opposing functions at a high level, such as the accelerator and brake, in a sports car.

This experiment was conducted at the trial production stage of the new tape; therefore, the number of samples was small. Evaluating a more significant number of samples is necessary when concrete commercialization is in sight. Furthermore, we must develop our prototype to ensure test safety before conducting a peeling test on older adults with fragile skin. The participants were instructed to close their eyes when removing the tape, and the order of tape removal was randomized; however, complete blindness could not be achieved during the experiment.

Finally, the new surgical tape with mesh does not have high adhesive strength. Moreover, the mesh caused prickly skin irritation and reticulated skin deformation after application (although they recovered without problems after some minutes). It should also be noted that part of the mesh stuck to the adhesive and stretched slightly while removing the tape (Fig 5 and S2 Video). Therefore, it is necessary to determine the best combination of tape substrate, mesh, adhesive, and product design in the future. By using a thin and flexible film as the tape substrate and improving the tape's ability to conform to the skin, it may be possible to obtain strong adhesion even with mesh and reduce itchy skin irritation caused by the mesh. Improvements in such as biocompatibility, manufacturing cost, tape design that anyone can use are expected to lead to products that surpasses silicon taping and the work of peeling off with a remover in clinical site. Improvements in such as biocompatibility, manufacturing cost, tape design that anyone can use are expected to lead to products that surpasses silicon taping and the work of peeling off with a remover in clinical site. If an appropriate combination is determined, we believe that the limited use of mesh tape will be possible for the fixation of drips in extremities, protection of drip insertion sites, fixation of intubation tubes, and skin care of patients with bullous disease.

## Conclusion

The new type of surgical tape with mesh designed to prevent skin tears reduced the sensation of skin pain during tape removal from human skin. With conventional tape, there is a trade-off between firmly fixing the medical device with the tape and gently removing it because a pressure-sensitive adhesive must have both tape-fixing strength and ease of tape separation. With this new tape, the mesh protects the skin, and the strong adhesive firmly grips the skin, thereby resolving this problem. We will develop a thin and strong mesh, make the tape-based material transparent and waterproof, and add a device that allows the mesh to be smoothly peeled off from the tape-based material. We would like to conduct further experiments to demonstrate the new tape's stronger adhesion and skin protection.

## Supporting information

**S1 Fig. Schema of deflection due to gravity on a beam.**
(TIF)

**S1 Video. The skin condition when the tape with mesh was peeled off together.**
(MP4)

**S2 Video. The skin condition when the tape with mesh was peeled off while leaving the mesh on the skin.**
(MP4)

## Acknowledgments

The authors thank Editage (www.editage.com) for English language editing.

## Author Contributions

**Conceptualization:** Hiroaki Rikhisa, Masayuki Nakao.

**Data curation:** Naoaki Rikihisa.

**Formal analysis:** Naoaki Rikihisa.

**Funding acquisition:** Naoaki Rikihisa, Hiroaki Rikhisa.

**Investigation:** Nobuyuki Mitsukawa.

**Methodology:** Nobuyuki Mitsukawa.

**Project administration:** Nobuyuki Mitsukawa.

**Validation:** Masayuki Nakao.

**Writing – original draft:** Naoaki Rikihisa.

**Writing – review & editing:** Nobuyuki Mitsukawa, Hiroaki Rikhisa, Masayuki Nakao.

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
