## [Decision Letter · Decision Letter 0]

6 Apr 2023

PONE-D-22-33348A new surgical tape with a mesh designed to prevent skin tears and reduce pain during tape removalPLOS ONE

Dear Dr. Rikihisa,

Thank you for submitting your manuscript to PLOS ONE. After careful consideration, we feel that it has merit but does not fully meet PLOS ONE’s publication criteria as it currently stands. Therefore, we invite you to submit a revised version of the manuscript that addresses the points raised during the review process.

We look forward to receiving your revised manuscript.

Kind regards,

Jianhong Zhou

Staff Editor

PLOS ONE

Journal Requirements:

"We have read the journal's policy and the authors of this manuscript have the following competing interests:

Hiroaki Rikhisa is a holder of the patents related to the surgical tape with the mesh.

This study supported by Entrepreneurs Program of New Energy and Industrial Technology Development Organization (22100741-0)."

Reviewers' comments:

Reviewer's Responses to Questions

**Comments to the Author**

1. Is the manuscript technically sound, and do the data support the conclusions?

Reviewer #1: Partly

Reviewer #2: Yes

Reviewer #3: Partly

2. Has the statistical analysis been performed appropriately and rigorously? 

Reviewer #1: N/A

Reviewer #2: Yes

Reviewer #3: I Don't Know

3. Have the authors made all data underlying the findings in their manuscript fully available?

Reviewer #1: Yes

Reviewer #2: Yes

Reviewer #3: No

4. Is the manuscript presented in an intelligible fashion and written in standard English?

Reviewer #1: Yes

Reviewer #2: Yes

Reviewer #3: Yes

5. Review Comments to the Author

Reviewer #1: The article “A new surgical tape with a mesh designed to prevent skin tears and reduce pain during tape removal” has provided emphasized on considerable topic. The surgical tapes are very painful when taken off after the surgery. This seems to be very innovative idea to put mesh in between the skin and the tape.

Here are some questions for the authors;

1. Did the authors find any kind of side effect of mesh on the skin?

2. Does this tape removal need any expertise or any other person can do it?

3. Does this tape is cost effective?

4. Did authors compared this method with the other different approaches (under clinical trial) for preventing medical adhesive-related skin injury?

Reviewer #2: The manuscript is technically sound and all the data mentioned in the manuscript are supporting the conclusions. Statistical data are analyzed appropriately and clearly. All the data mentioned in the manuscript are available in the literature. The manuscript is presented in an intelligent fashion and written in standards English. The correction may kindly be made as per the reviewed manuscript.

Reviewer #3: 1. The description of new tape in terms of the materials and what properties been considered for its selection for study need to be thoroughly introduced in the Introduction.

2. Details of the substrate and adhesive used in the study need to be more elaborated

3. The manuscript is lacking tables and figures

4. Statistical data and the results are not provided

5. Language is scientific with clarity of expression

6. Please make sure that words are separated by a space

7. Reference line 363 author name has been corrected 'Lamers E, van Kempen THS, Baaijens, FPT, Peters GWM, Oomens CWJ'

6. PLOS authors have the option to publish the peer review history of their article (what does this mean?). If published, this will include your full peer review and any attached files.

Reviewer #1: **Yes: **DEEPTI SHARMA

Reviewer #2: **Yes: **Dr. Bedanga Konwar

Reviewer #3: No

---

## [Author Response · Author response to Decision Letter 0]

26 Apr 2023

POINT-BY-POINT RESPONSES TO REVIEWERS

This does not alter our adherence to PLOS ONE policies on sharing data and materials.

REVIEWER #1:

(Comment 1)

The article “A new surgical tape with a mesh designed to prevent skin tears and reduce pain during tape removal” has provided emphasized on considerable topic. The surgical tapes are very painful when taken off after the surgery. This seems to be very innovative idea to put mesh in between the skin and the tape. Here are some questions for the authors.

(Response) Thank you for pointing out the problem of pain during tape removal. And thank you very much for scrutinizing our manuscript. According to your kind suggestions, we revised the manuscript. They were very helpful to improve its quality. We would appreciate your re-consideration of our revised paper.

(Comment 2)

Did the authors find any kind of side effect of mesh on the skin?

(Response)

Thank you very much for important question about side effect of mesh on the skin. Our mesh is made from olefin and was made to be as thin as possible while still maintaining its stiffness. But the mesh caused prickly skin irritation and reticulated skin deformation after application (although they recovered without problems after some minutes). (in the Discussion section on page 15, lines 333 to 334)

We are still working on improving the tape. Changing the base material from non-woven fabric to film improved the tape's ability to conform to the skin. As a result, there is less fine movement of the mesh between the skin and the base material, reducing skin irritation.

Therefore, we added the following sentences in the Discussion section on page 16, lines 337 to 340: “By using a thin and flexible film as the tape substrate and improving the tape's ability to conform to the skin, it may be possible to obtain strong adhesion even with mesh and reduce itchy skin irritation caused by the mesh.” 

(Comment 3)

Does this tape removal need any expertise or any other person can do it?

(Response)

Thank you for pointing out the tape usability that is important task in the developing new tape. The product should be designed so that even nurses with presbyopia can easily remove the tape. We have added the following sentences in the Discussion section on page 16 lines 340 to 343: “Improvements in such as biocompatibility, manufacturing cost, tape design that anyone can use are expected to lead to products that surpasses silicon taping and the work of peeling off with a remover in clinical site.”

(Comment 4)

Does this tape is cost effective?

(Response)

Thank you for the question about the production cost. We are developing the new tape that is cheaper than or equal to silicon tape. We have added the following sentences in the Discussion section on page 16, lines 340 to 343: “Improvements in such as biocompatibility, manufacturing cost, tape design that anyone can use are expected to lead to products that surpasses silicon taping and the work of peeling off with a remover in clinical site.”

(Comment 5)

Did authors compared this method with the other different approaches (under clinical trial) for preventing medical adhesive-related skin injury?

(Response)

We are sorry that we have not yet performed a comparative analysis of other approaches and our mesh method for preventing medical adhesive-related skin injury. Each method probably has advantages and disadvantages, and it may be difficult to scientifically judge their superiority. 

REVIEWER #2:

(Comment 1)

The manuscript is technically sound and all the data mentioned in the manuscript are supporting the conclusions. Statistical data are analyzed appropriately and clearly. All the data mentioned in the manuscript are available in the literature. The manuscript is presented in an intelligent fashion and written in standards English. The correction may kindly be made as per the reviewed manuscript.

(Response)

Thank you for your positive comments on our manuscript. We feel honored.

REVIEWER #3:

(Comment 1)

The description of new tape in terms of the materials and what properties been considered for its selection for study need to be thoroughly introduced in the Introduction.

(Response)

Thank you very much for scrutinizing our manuscript. According to your important questions and suggestions, we revised the manuscript. 

We have added the following sentences to introduce the experiment tape in the Introduction section on page 3, lines 58 to 62: “A tape manufacturer combined mesh with the medical tape used to secure dialysis routes in hospitals to create the experimental tape. We attempted to clarify the advantages and disadvantages of mesh by adding no new components of medical tape other than mesh in this experiment.”

(Comment 2)

Details of the substrate and adhesive used in the study need to be more elaborated.

(Response)

Thank you for pointing out the lack of information about the experimental tapes. Since the properties of mesh has already been described in the manuscript, the description of the tape substrate, the adhesive, and their processing method has been added in the Material and Methods section on page 4, line 75 to 79: “Kino white (Nitoms, Inc., Tokyo, Japan) was used as the tape substrate and the adhesive of the experimental tape. Kino white is a highly adhesive medical tape composed of non-woven fabric and acrylic adhesive, and is used in dialysis hospitals to fix tubes to the patient's skin. The experimental tape was made by a tape manufacturer (Cosmotec Co., Ltd.) by combining Kino white and olefin mesh.”

(Comment 3)

The manuscript is lacking tables and figures.

(Response)

Thank you for reviewing our manuscript despite the lack of figures. The figures were submitted to the editors on the web. Sorry for the inconvenience, we kindly ask to request figures from the editor.

(Comment 4)

Statistical data and the results are not provided.

(Response)

Thank you for your comment. The results of the statistical analysis as figures 6, 7, and 8 have been submitted to the editor on the web. We kindly ask to request the figures from the editor. And we will additionally submit measurement results of Degree of pain as a supplemental table, so please check and evaluate the table.

(Comment 5)

Language is scientific with clarity of expression.

(Response)

Thank you for your comment.

(Comment 6)

Please make sure that words are separated by a space.

(Response)

Thank you for pointing out our mistakes. We made several corrections as below points: Page 2, line 33, 35, 39., Page 6, line 115, 127., Page 7, line 132., Page 8, line 169., Page 10, line 214., Page 11, line 223., Page14, line 308., Page 19, line 401.

(Comment 7)

Reference line 363 author name has been corrected 'Lamers E, van Kempen THS, Baaijens, FPT, Peters GWM, Oomens CWJ'.

(Response) 

Thank you for pointing out the mistake. We made corrections as below points in the References section on page 17, line 371 to 373: “1. Lamers E, van Kempen THS, Baaijens FPT, Peters GWM, Oomens CWJ. Large amplitude oscillatory shear properties of human skin. J Mech Behav Biomed Mater. 2013;28: 462-470. doi:10.1016/j.jmbbm.2013.01.024.”

---

## [Decision Letter · Decision Letter 1]

26 Jun 2023

A new surgical tape with a mesh designed to prevent skin tears and reduce pain during tape removal

PONE-D-22-33348R1

Dear Dr. **Naoaki Rikihisa**

We’re pleased to inform you that your manuscript has been judged scientifically suitable for publication and will be formally accepted for publication once it meets all outstanding technical requirements.

Kind regards,

Sakar Palecha, Ph.D.

Guest Editor

PLOS ONE

Additional Editor Comments (optional):

Reviewers' comments:

Reviewer's Responses to Questions

**Comments to the Author**

1. If the authors have adequately addressed your comments raised in a previous round of review and you feel that this manuscript is now acceptable for publication, you may indicate that here to bypass the “Comments to the Author” section, enter your conflict of interest statement in the “Confidential to Editor” section, and submit your "Accept" recommendation.

Reviewer #1: All comments have been addressed

Reviewer #2: All comments have been addressed

2. Is the manuscript technically sound, and do the data support the conclusions?

Reviewer #1: Yes

Reviewer #2: Yes

3. Has the statistical analysis been performed appropriately and rigorously? 

Reviewer #1: Yes

Reviewer #2: Yes

4. Have the authors made all data underlying the findings in their manuscript fully available?

Reviewer #1: Yes

Reviewer #2: Yes

5. Is the manuscript presented in an intelligible fashion and written in standard English?

Reviewer #1: Yes

Reviewer #2: Yes

6. Review Comments to the Author

Reviewer #1: The authors of the article “A new surgical tape with a mesh designed to prevent skin tears and reduce pain during tape removal” has included all the comments raised in the manuscript and explained them properly.

The manuscript can be considered for the publication in “PLOS One”.

Reviewer #2: The manuscript may be accepted for publication after looking to the reviewer comment. The manuscript is technically sound. Little modification may be done

7. PLOS authors have the option to publish the peer review history of their article (what does this mean?). If published, this will include your full peer review and any attached files.

Reviewer #1: **Yes: **DEEPTI SHARMA

Reviewer #2: **Yes: **Bedanga Konwar

---

## [Editor Report · Acceptance letter]

29 Jun 2023

PONE-D-22-33348R1 

A new surgical tape with a mesh designed to prevent skin tears and reduce pain during tape removal 

Dear Dr. Rikihisa:

I'm pleased to inform you that your manuscript has been deemed suitable for publication in PLOS ONE. Congratulations! Your manuscript is now with our production department. 

Kind regards, 

on behalf of

Dr. Sakar Palecha 

Guest Editor

PLOS ONE